# Extracellular Vesicles in Myeloid Neoplasms

**DOI:** 10.3390/ijms23158827

**Published:** 2022-08-08

**Authors:** Christina Karantanou, Valentina René Minciacchi, Theodoros Karantanos

**Affiliations:** 1Georg-Speyer-Haus, Institute for Tumor Biology and Experimental Therapy, 60596 Frankfurt am Main, Germany; 2Division of Hematologic Malignancies and Bone Marrow Transplantation, Sidney Kimmel Comprehensive Cancer Center, Johns Hopkins University, Baltimore, MD 21218, USA; 3The Sidney Kimmel Comprehensive Cancer Center, Johns Hopkins University School of Medicine, The Bunting-Blaustein Cancer Research Building, 1650 Orleans Street, Baltimore, MD 21218, USA

**Keywords:** extracellular vesicles, myeloid malignancies

## Abstract

Myeloid neoplasms arise from malignant primitive cells, which exhibit growth advantage within the bone marrow microenvironment (BMM). The interaction between these malignant cells and BMM cells is critical for the progression of these diseases. Extracellular vesicles (EVs) are lipid bound vesicles secreted into the extracellular space and involved in intercellular communication. Recent studies have described RNA and protein alterations in EVs isolated from myeloid neoplasm patients compared to healthy controls. The altered expression of various micro-RNAs is the best-described feature of EVs of these patients. Some of these micro-RNAs induce growth-related pathways such as AKT/mTOR and promote the acquisition of stem cell-like features by malignant cells. Another well-described characteristic of EVs in myeloid neoplasms is their ability to suppress healthy hematopoiesis either via direct effect on healthy CD34+ cells or via alteration of the differentiation of BMM cells. These results support a role of EVs in the pathogenesis of myeloid neoplasms. mainly through mediating the interaction between malignant and BMM cells, and warrant further study to better understand their biology. In this review, we describe the reported alterations of EV composition in myeloid neoplasms and the recent discoveries supporting their involvement in the development and progression of these diseases.

## 1. Introduction

Myeloid neoplasms are clonal diseases arising from malignant stem and progenitor cells that have acquired somatic mutations in genes regulating hematopoiesis, cell differentiation, apoptosis and proliferation [1]. These cells exhibit survival and growth advantage within the bone marrow (BM) microenvironment against healthy hematopoietic stem and progenitor cells which can lead to accelerated growth, clonal evolution and disease progression [2]. A critical component in the pathogenesis and progression of these diseases is the interaction of malignant cells with the BM microenvironment [3]. Numerous studies have highlighted that the presence of malignant myeloid cells induces alterations in the transcriptional and proteomic profile of the BM stromal cells with important implications in their function [3,4]. These functional changes of the stromal cells create a BM niche that protects the malignant cells and suppresses healthy hematopoiesis mediating disease progression [3,5,6]. Better understanding of the exact molecular mechanisms implicated in this crosstalk between malignant cells and BM stromal cells is required for the development of more effective targeted therapies.

Extracellular vesicles (EVs) have been suggested to play an important role in cell–cell communication by mediating the exchange of complex information [7]. These particles are released by cells both in physiological and pathological conditions and carry several types of macromolecules, nucleic acids, proteins and lipids across the extracellular milieu to target cells [7]. Recent data have highlighted the critical role of EVs in the crosstalk between malignant cells and cells of their microenvironment such as endothelial, mesenchymal and immune cells [8,9]. In particular, studies have shown that secretion of EVs from malignant cells is critical for the re-programming of the tumor microenvironment towards a more protective niche that supports the malignant cells’ survival, proliferation, immune escape and resistance to cytotoxic therapies [9,10]. On the other hand, stromal cells promote malignant cells’ survival and proliferation, inhibit immune response and suppress the survival of healthy cells via secretion of EVs in the tumor microenvironment [10,11,12].

The aim of this review is to summarize the current literature on the role of EVs in the development and progression of myeloid neoplasms with emphasis on the crosstalk between malignant myeloid cells and the BM microenvironment.

## 2. Extracellular Vesicles

EVs are lipid bound vesicles secreted by cells into the extracellular space [13]. Several studies have pointed out the great heterogeneity that lies behind what are called EVs [14,15]. Different parameters, such as sub-cellular origin, size and composition, have been used to categorize the different subpopulations of EVs. Based on the size, four main EV populations are defined [16,17,18] (Figure 1): Exosomes (~50–150 nm), which are released from several cells both in physiological and pathological conditions; micro-vesicles (100–1000 nm), shed from normal and transformed cells; apoptotic bodies (100–5000 nm) which result from the fragmentation of dying apoptotic cells; and large oncosomes (1000–10,000 nm) that have been demonstrated to shed from tumor cells with high membrane plasticity. Various studies aimed at investigating the composition of these EV populations have shown the sorting of specific molecules into distinct EVs. This enrichment can be distinguished into two types: 1. specific to the EV population which is often associated with EV biogenesis and used to determine the identity of a certain subpopulation; 2. specific to the donor cell. The last has prompted a number of studies aimed at addressing the employment of EVs as source of biomarkers to follow up disease progression and/or treatment response [19,20]. One such example is the identification of a melanoma cell-derived exosomal signature consisting of chaperon proteins and oncoproteins that showed both prognostic and therapeutic potential [21], or the discovery of unique phosphoproteins and phospho-peptides in patients diagnosed with breast cancer compared to healthy controls [22]. Additionally, the detection of high levels of exosomes with specific signatures in the circulation of cancer patients, such as breast [23] and pancreatic cancer [24], have also been associated with the development of metastasis, rendering them important predictive and/or therapeutic biomarkers. Finally, in the case of prostate cancer a non-invasive urine exosome gene expression assay has been suggested to discriminate high-grade from low-grade cancer and benign disease, reducing the need for unnecessary urinary biopsies [25].

For their ability to transfer information across cells, EVs have been the object of several studies aimed at investigating the mechanisms underlying their trafficking, biogenesis and uptake, and the effects of EVs once they interact with target cells [17,18]. This is important information that might add to our knowledge of how cells interact and affect each other both in physiological and pathological conditions within a specific environment, paracrine and systemic communication. Particularly, in pathological conditions like cancer, a better understanding of how EVs are exchanged between malignant cells and their microenvironment gives information on how, on the one hand, tumor cells educate the surrounding cells and, on the other hand, how the different cells in the microenvironment support the survival and growth of malignant cells [10,17]. Furthermore, unveiling the mechanisms behind EV biogenesis, EVs’ interaction with specific target cells and, ultimately, EV uptake would allow the development of approaches to interfere with the EV based communication that contributes to disease progression [26].

To date two main subcellular sites of origin have been proposed: the endosomal system and the plasma membrane. Exosomes have been mainly described as originating as intraluminal vesicles within the multivesicular bodies (MVBs) and then released upon fusion of the MVBs with the plasma membrane. Several regulators of this mechanism have been so far identified and these include the neutral sphingomyelinase 2 and ceramide regulation [27,28,29] components of the endosomal sorting complex [30,31] RAB proteins [32,33,34] and regulators of actin polymerization [35]. Interestingly glyceraldehyde-3-phosphate dehydrogenase (GAPDH) has been recently involved in EV biogenesis through its activity in regulating the formation of intraluminal vesicles [36]. Plasma membrane budding has been implicated in the biogenesis of multiple type of EVs including micro-vesicles and large EVs [37,38]. A key step in this process is the onward budding of the plasma membrane, which has often been associated with cytoskeletal rearrangements and increased membrane deformability [39,40,41].

Once in the extracellular milieu, EVs have been suggested to adopt different routes in order to enter the target cells. These include macro-pinocytosis, clathrin/caveolin-mediated endocytosis, phagocytosis, and membrane fusion [42]. However, it is still not clear what determines the decision favoring one mechanism rather than another. One contributing factor could represent the size of the EV that is entering the cell. Another could be represented by the presence or absence on the surface of specific molecules that can either determine the specific target cell [43] or initialize the signaling cascade that leads to the internalization of EVs [44,45,46]. To note, it has also been shown that EVs can affect target cells by membrane interaction without entering the target cells [47], adding another layer of complexity to the EV-mediated effects on target cells.

Unfortunately, uptake does not necessarily translate into successful delivery of EV-cargo and, despite the progress in understanding how EVs enter the cells, the fate of these EVs and their cargo remains unknown. Recent studies have attempted to analyze what happens to EV-associated molecules inside the target cells, employing strategies that allow the monitoring of EV-cargo [48,49,50]. Notably, one appealing approach in this direction is to translate to EV research knowledge on the mechanisms adopted by viruses.

Understanding EV biology offers several therapeutic opportunities which not only include the possibility of affecting and interfering with the intercellular communication mediated by E,Vs but also the possibility of altering EV composition and using them to deliver specific cargoes [51,52]. Furthermore, knowing the mechanism of uptake can also help in developing strategies to engineer EVs for cargo delivery such as the development of a macro-pinocytosis-inducing peptide [53] or alteration of glycan composition (glycoengineering) [54].

### 2.1. Extracellular Vesicles in Philadelphia-Negative Myeloproliferative Neoplasms

Philadelphia-negative myeloproliferative neoplasms (MPN) are a group of chronic myeloid neoplasms consisting of essential thrombocytosis (ET), polycythemia vera (PV) and primary myelofibrosis (PMF) [55,56]. These neoplasms arise from malignant stem cells carrying mutations that lead to constitutively active JAK2/STAT signaling and provide them with survival and growth benefits against healthy hematopoietic cells [57]. ET is characterized by markedly elevated platelet counts, higher incidence of thrombosis, but overall similar survival to the general population [58]. PV is also a relatively indolent myeloid neoplasm characterized by elevated red blood cell production. Despite that most PV patients have a survival that is comparable to the general population, the incidence of disease progression to secondary myelofibrosis and acute leukemia within 20 years from diagnosis is 16% and 4%, respectively [59]. Finally, PMF is a myeloid neoplasm driven by the same stem cell mutations but with a much more aggressive natural history including extensive bone marrow fibrosis, failure of normal hematopoiesis, extramedullary hematopoiesis, increased risk of leukemic transformation and overall poor survival [60].

It is well described that the presence of MPN malignant stem and primitive cells causes significant alterations in BM microenvironment cells, creating a niche which protects and re-enforces the survival and growth of the malignant cells [61]. Within this marrow niche, the release of high levels of inflammatory cytokines such as tumor necrosis factor α (TNFα) and tumor growth factor β1 (TGFβ1) contribute to the dominance of malignant clones at the expense of the normal hematopoietic cells [62] and is associated with the increased deposition of reticulin and collagen fibers leading to the development of extensive marrow fibrosis, a common finding for MPN patients [63]. Similarly, *chemokine (C-X-C motif) ligand 4* (*CXCL4*), which is expressed in malignant hematopoietic cells, codes for a protein that can activate stromal cells and control BM fibrosis in patients with MPN, [64] highlighting that the crosstalk between malignant cells and the BM microenvironment via cytokine signaling is critical for the pathogenesis of these neoplasms.

Apart from cytokines and soluble factors, malignant cells and components of the BM microenvironment communicate via the reciprocal exchange of EVs, which carry bioactive molecules, such as proteins, including proteins embedded in the EV membrane-like receptors, nucleic acid molecules and lipids that can alter cell behavior, in terms of proliferation, adhesion and survival upon uptake [65,66]. The role of EVs has been studied intensively as part of the microenvironment of solid tumors but less is known of their involvement in hematological malignancies. However, there is increasing evidence supporting a key role of EVs in the progression of myeloid neoplasms via their involvement in inflammation and immunomodulation, and particularly MPN [67,68].

### 2.2. Extracellular Vesicles in Essential Thrombocytosis

ET patients have elevated levels of platelet-derived microparticles, a sub-type of EVs generated from the plasma membrane of platelets upon their activation by various stimuli [69]. Of note, elevated CD41+ microparticles were positively associated with increased thrombosis risk among ET patients [70]. In a follow up study on the mechanism of this association by the same group, using a thrombin generation assay increased generation of thrombin in patients with ET compared to healthy controls was demonstrated, but importantly, # also in *JAK2*-V617F positive compared to *JAK2*-V617F negative patients, implicating the role of microparticles [71]. Thus, it is possible that the pro-thrombotic effect of *JAK2* V617F mutation in ET is partially mediated by the secretion of platelet-derived EVs with thrombin generation activity. Consistently, it has been showed that anagrelide which inhibits the maturation of megakaryocytes to platelets and is commonly used to treat ET patients can decrease the levels of circulating EV back to normal among these patients [72].

Exosomes are among the smallest of the EVs, and undergo a complex process that involves inward budding of endosomes [73]. Consistent with the presence of only a few papers on the role of EVs in essential thrombocythemia, the role of exosomes has, to our knowledge, only been addressed in one study where the analysis of BM-derived exosomes from ET patients has revealed an altered non-coding RNA profile compared to healthy individuals, with significant downregulation of the circDAP3, circASXL1, and circRUNX1 circular RNAs [74]. Importantly, circular RNA derived from the exosomes of ET patients was implicated in cellular processes such as proliferation and apoptosis and was found to inhibit the maturation of K562 cells to megakaryocytes. Suppression of differentiation is a biological process that contributes to the progression of MPNs to more advance stages such as myelofibrosis and acute leukemia. These results further highlight the importance of EV cargo in the development and progression of MPN as it is possible that through the secretion of EVs malignant MPN cells can affect critical biological functions of other hematopoietic cells within the BM niche. The reported findings related to the role of EVs in ET are summarized in Table 1.

### 2.3. Extracellular Vesicles in Polycythemia Vera

It has been reported that PV patients, similarly to ET patients, have elevated levels of EVs of platelet origin in their plasma compared to healthy controls [75]. Comparison of the serum EV-enriched proteome between PV patients and healthy controls revealed that PV patients express significantly higher levels of proteins associated with platelet activation, induced immune and inflammatory responses, coagulation and angiogenesis [76]. More recently, Barone et al. confirmed the increased levels of platelet derived EVs in the blood of PV patients compared to healthy individuals and demonstrated increased diversity and a different microbial DNA signature of the released EVs in PV patients [77]. Of note, EVs secreted by platelets have been found to harbor significant pro-coagulant activity [78] while venous and arterial thrombosis is the major source of morbidity and mortality among PV patients [57,79]. Given that in these studies the responsible molecules through which these EVs express their pro-coagulant activity are not yet identified, it is difficult to draw conclusions on the mechanism by which this is achieved, as well as on whether this mechanism differs compared to the one involved in the ET. Despite the small amount of evidence, one may conclude that EVs released by malignant MPN clones could be implicated in the pathogenesis of thrombosis in MPN, representing a promising target for new therapeutic strategies. The reported findings related to the role of EVs in PV are summarized in Table 1.

### 2.4. Extracellular Vesicles in Primary Myelofibrosis

The natural history of PMF is characterized by particularly altered cytokine profile in the BM microenvironment, which is directly linked with the excess marrow fibrosis and bone marrow failure [80]. EVs are involved in the intercellular communication through their release in the extracellular space of the bone marrow microenvironment and may regulate the progression of some hematological malignancies like multiple myeloma [81]. Even though the observations in PMF are not focused on the BM microenvironment itself, it was observed that circulating EVs, specifically lower than 0.3 μm micro-vesicles, are significantly elevated in patients with PMF compared to healthy controls [82]. Among MPN patients, patients with PMF have significantly higher levels of platelet-, endothelial cell- and erythrocyte-derived microparticles compared to patients with PV [83] and significantly higher levels of erythrocyte-derived microparticles compared to patients with ET [83]. Interestingly, patients with PMF have decreased levels of circulating megakaryocyte-derived micro-vesicles but increased levels of circulating platelet-derived micro-vesicles compared to healthy controls [84], which may reflect the defective megakaryocytic activity and thrombopoiesis in PMF. Consistently, splenomegaly and thrombocytopenia were negatively correlated with the megakaryocyte-derived micro-vesicles among PMF patients [84]. Finally, treatment with Ruxolitinib, a JAK1/2 inhibitor, which is the most commonly used treatment for PMF patients, led to an increase in the megakaryocyte-derived micro-vesicles, rendering them a potential biomarker of disease activity and response to therapy [84].

About 10–15% of PMF patients do not have a mutation in one of the three known genes (*JAK2*, *CALR*, or *MPL*) and their disease is called triple negative (TN) [57]. These patients show significantly worse survival compared to PMF patients with a detectable mutation [57]. EVs isolated from the plasma of patients with TN disease increased the survival of CD34^+^ cells from the same patients in in vitro co-culture experiments, which was not the case for PMF patients with *JAK2* mutation or healthy individuals. Analysis of the miRNA profile of the isolated EVs revealed miR-361-5p to be the only different molecule in TN PMF patients compared to the other two groups, but no further functional experiments were performed to give further information on the exact mechanism [85]. This is an important observation given that cancer stem cell-derived EVs are proved to propagate cancer stem cells, for example, through the promotion of stem-like characteristics in non-cancer stem cells, or the modulation of the environment [86]. However, further studies need to be conducted to understand if and how this is applied in the case of TN PMF.

One of the main characteristics of PMF is the significant impairment of healthy hematopoiesis and normal white blood cell differentiation, which is associated with severe immunodeficiency [60]. In a study investigating the role of the immune system and inflammation in PMF patients, especially after treatment with ruxolitinib, the importance of EV-linked cytokines after infection was demonstrated. Specifically, infection, represented by lipopolysaccharides (LPS) stimulation, impaired the release of both free and EV-linked cytokines by monocytes in PMF patients with *JAK2*V617F mutation, while ruxolitinib restored only the EV-linked and not the free cytokines. Given that infection represents the cause for 10% of patients with myelofibrosis [87] these results may shed light on the underlying mechanism and its potential targeting. The reported findings related to the role of EVs in PMF are summarized in Table 1.

Overall, these results highlight that EV secretion is altered in PMF patients at a higher extent compared to the more indolent ET and PV, reflecting the markedly altered BM-niche biology and dysfunctional hematopoiesis in this disease. Finally, EVs could represent promising targets for therapy and exciting novel biomarkers for disease progression and response to treatments. However, better understanding of their biology at the molecular level is required to solidify these associations and support their translational potential.

### 2.5. Extracellular Vesicles in Chronic Myeloid Leukemia

Chronic myeloid leukemia (CML) is a clonal myeloproliferative neoplasm derived from a fusion of the *Abelson murine leukemia (ABL1)* gene on chromosome 9 with the *breakpoint cluster region (BCR)* gene on chromosome 22 in primitive hematopoietic cells resulting in the expression of an oncoprotein called BCR-ABL1 leading to uncontrolled cell proliferation and suppression of apoptosis [88]. The introduction of first, second, third and now fourth generation of tyrosine kinase inhibitors (TKIs) in the clinic have tremendously improved the survival of CML patients even in cases of accelerated or blast phase [88,89].

However, it is believed that TKIs may not eliminate the dormant CML stem cells, which may result in disease relapse following treatment discontinuation [90]. Similarly, a small percentage of patients develop resistance to TKIs, usually through the acquisition of *BCR-ABL1* kinase domain mutations [91], or present with refractory disease requiring chemotherapy-based therapies followed by allogeneic bone marrow transplantation [88]. Finally, recent studies have highlighted the importance of the interaction of *BCR-ABL1* mutated cells with the BM microenvironment and particularly integrins [92], adhesion molecules [93], cytokines and cytokine receptors [94,95] and growth factors [96].

Thus, better understanding of the molecular biology of *BCR-ABL1* mutated stem cells growth, particularly under the influence of the BM microenvironment, is still required for the development of effective therapies for the cases of refractory disease or emergence of resistance to TKIs.

CML-derived exosomes promote the proliferation of CML cells in a direct autocrine manner but also indirectly via the BMM. It has been demonstrated that CML exosomes stimulate CML cell proliferation and colony formation via upregulation of anti-apoptotic molecules and downregulation of pro-apoptotic molecules [97]. These exosomes were found to be enriched in TGFβ1 while blockade of TGFβ1 signaling inhibited the exosome-mediated induction of CML cell proliferation [97]. Additionally, exosomes derived from CML cell lines or CML patients carried amphiregulin, an epidermal growth factor receptor (EGFR) ligand which stimulated the EGFR downstream signaling in BM stromal cells. Upon EGFR activation there was increased expression of IL8 and of the metalloproteinase MMP9, which in turn promoted the proliferation and survival of leukemia cells [98]. Furthermore, it was shown that exposure of the HS5 stromal cell line to CML-exosomes increased Annexin2 levels, thus, promoting the adhesion of CML cells on the stroma. This study is important as it implicates the involvement of CML-exosomes in the modulation of the environment through the interaction of a ligand carried by EVs and its associated receptors in the recipient cell [98]. Moreover, CML-exosomes can change the profile of cytokines such as TNFα, TGFβ1 and IL-10, the production of nitric oxide (NO) and the redox potential of BM mesenchymal stem cells and macrophages [99]. Importantly, these alterations transformed the mesenchymal stem cells and macrophages into leukemia-promoting cells through processes such as the polarization of macrophages to tumor-associated macrophages [99]. These findings support that CML-derived exosomes may be implicated in the re-programming of the BM microenvironment, creating a niche protecting CML cells.

CML cells induce angiogenesis implicating endothelial cells in the process of CML development [100]. Interestingly, it was shown that EVs released by CML cells carry the *BCR-ABL1* RNA and the BCR-ABL1 protein and can transfer both to endothelial cells but without direct link to altered cell phenotype [101]. It was also demonstrated that CML-derived exosomes stimulate tube formation and induce the angiogenic activity of HUVEC cells via upregulation of SRC phosphorylation, while dasatinib, a second generation TKI, inhibited the exosome production and vascular differentiation and signaling [102].

As a different mechanism, BM stromal cells can release exosomes, which can then influence leukemia cells. In CML, exosomes released by BM stromal cells transfer fibroblast growth factor 2 (FGF2) to CML cells, which protects them from the effect of TKIs [103]. This protective mechanism was reversed by FGFR inhibition, which also reduced exosome secretion [103]. The reported findings related to the role of EVs in CML are summarized in Table 1.

Overall, these results suggest a possible involvement of exosomes derived from CML or BM stromal cells in the development of a BM niche that protects CML cells supporting their survival, growth and resistance to TKIs. The elucidation of the molecular mechanisms of the regulation of exosomes production and secretion can lead to the discovery of new targeted therapies that can be used in combination with TKIs for patients with advanced or refractory CML.

### 2.6. Extracellular Vesicles in Myelodysplastic Syndrome

Myelodysplastic syndrome (MDS) is a relatively common myeloid neoplasm among elder individuals arising from hematopoietic stem cells that acquire somatic mutations, exhibit growth advantage in the BM microenvironment and fail to differentiate into normal white blood cells, erythrocytes and platelets [104]. MDS is a disease with particularly variable outcomes depending on genomic alterations, karyotypic abnormalities and severity of cytopenias [105]. Sex-related differences have been described in MDS with men having overall worse outcomes compared to women, associated with a more complex genomic profile [106,107,108]. Bone marrow failure and transformation to an aggressive form of acute leukemia called secondary acute myeloid leukemia are the main causes of death of patients with MDS [109]. Unfortunately, the therapeutic options of patients with high-risk MDS remain limited and their responses to these treatments are usually transient [104,109] with allogeneic bone marrow transplantation being the only curative approach for these individuals [109]. Thus, elucidation of the molecular mechanisms implicated in the growth of MDS cells in the BM microenvironment is required for the development of novel therapies that will improve the survival of high-risk MDS patients.

It has been well described that MDS cells induce specific alterations in the BM microenvironment [110,111] transforming it to an “MDS-promoting” niche that in turn supports the growth and proliferation of malignant cells via the secretion of cytokines, growth factors and modulation of immune response [5,6,112,113,114] while suppressing normal hematopoiesis [111]. Similarly, MDS cells overexpress various cytokine receptors and co-receptors that mediate the activity of cytokines secreted by BM stromal cells, promoting the growth and suppressing the differentiation of malignant cells [115,116]. Thus, it is evident that a close interaction between BM stromal cells and MDS cells is required for the development and progression of this disease.

A study analyzing the expression profile of circulating small noncoding RNAs from MDS patients and healthy controls revealed that the expression of various hematopoiesis-related micro-RNA (miR) molecules is altered in the EV cargo of MDS patients [117]. Interestingly, the expression of four miR molecules (miR-1237-3p, U33, hsa_piR_019420, and miR-548av-5p) was negatively associated with the survival of MDS patients [117]. Similarly, the analysis of the proteasome of EV-rich fraction of plasma from 36 MDS patients and 12 healthy controls revealed that the expression of a number of proteins is different between high-risk MDS patients and healthy individuals with alterations in clusterin being the most prominent [117]. Thus, it is possible that MDS-derived EVs may affect other cells in the BM micro-environment. Indeed, it was recently demonstrated that MDS-derived EVs suppress the osteo-lineage differentiation of mesenchymal stromal cells, which impairs their ability to support healthy hematopoietic stem cells (HSC) [111]. Further analysis of the effect of MDS-derived EVs in the BM microenvironment cells and mechanistic studies are required to improve our understanding of their role in the interaction between MDS and BM niche.

Interestingly, apart from MDS-derived EVs, EVs from re-programmed BM stroma derived from MDS patients also present transcriptional alterations, and have been found to induce MDS-related changes and disease progression. Particularly, the expression of a number of miR such as miR-10a and miR-132 is altered in EVs from mesenchymal stromal cells derived from MDS patients compared to EVs from mesenchymal stromal cells derived from healthy individuals [118]. The authors also showed that these EVs could be taken up by healthy CD34+ cells inducing alterations in the expression of *MDM2* and *TP53* in these cells and in their clonogenicity [118]. It was also reported that the expression of miR-101 was downregulated in EVs derived from mesenchymal stromal cells from patients with high-risk MDS and acute myeloid leukemia compared to patients with low-risk disease [119]. Of note, miR-101 suppresses cell proliferation [119] suggesting that its downregulation could be implicated in the acceleration of MDS cell proliferation and transformation to acute leukemia. Finally, Meunier et al. recently found that small EVs from mesenchymal stromal cells derived from MDS patients induce ROS production promoting DNA damage and mutagenesis in healthy HSC via miRNA transfer [120]. Thus, the re-programmed MDS stromal cells secrete EVs that may induce alterations in both malignant and healthy cells that, overall, induce the progression of MDS. The reported findings related to the role of EVs in MDS are summarized in Table 1.

Overall, EVs secreted by MDS cells and mesenchymal stromal cells in the MDS BM microenvironment show significant alterations in the transcriptional and protein levels and may have important implications in not only the interaction between MDS cells and stromal cells but also in the suppression of healthy hematopoiesis. If these findings are further supported by additional studies with mechanistic insight into these associations, targeting EVs could be a promising therapeutic approach for high-risk MDS.

### 2.7. Extracellular Vesicles in Acute Myeloid Leukemia

Acute myeloid leukemia (AML) is the most common type of acute leukemia in adults having an incidence of 4.3 per 100,000 annually in the United States [121]. It arises from clonal expansion and sub-clonal evolution of malignant stem or progenitor cells and is characterized by significant biologic heterogeneity, which is reflected to the outcomes of AML patients [122]. Despite the improvement of our understanding of this disease biology and the introduction of numerous novel targeted therapies such as gemtuzumab ozogamicin, venetoclax, FLT3 inhibitors (midostaurin, gilteritinib), IDH inhibitors (ivosidenib, enasidenib), CPX-351, glasdegib, the 5-year survival rate of patients with this disease remains lower than 50% [122,123]. These overall poor survival outcomes are attributed to the very high relapse rates of patients with AML, which are associated with the presence of malignant stem cells, called leukemia stem cells (LSCs) that are resistant to chemotherapy and occasionally BM transplantation [124]. The interactions of LSCs with stromal cells in the BM niche are critical for their survival and their protection from traditional therapies and novel targeted therapies [125,126,127]. Thus, exploiting the role of EVs in AML especially in the context of this interaction of LSCs with their BM microenvironment is particularly interesting from the perspective of disease biology and the development of novel therapies.

Tumor-derived EVs have the potential to modulate the immune system, for example, by inducing apoptosis of effector T cells, upregulating the suppressive activity of regulatory T cells and suppressing NK cells activity [128]. One of the mechanisms associated with the survival of LSCs is the suppression of immune cells activity and particularly the downregulation of NK cells activating receptors [129]. Interestingly, micro-vesicles from the serum of AML patients decreased the cytotoxicity of NK cells and downregulated the expression of the activating NK receptor NKG2D [130]. This downregulation was mediated by TGFβ1, which was upregulated in these AML-derived exosomes [130].

Apart from the immune system, AML cells interact with and reprogram other aspects of the BM niche. AML cells release exosomes which carry elevated levels of AML-relevant RNA transcripts like insulin-like growth factor 1 (IGF)-I receptor, which upon uptake by stromal cells increase their proliferation capacity and vascular endothelial growth factor (VEGF) expression [131]. This indicates that transcripts transferred by AML exosomes have the potential to change the recipient stromal cell behavior. Moreover, the presence of AML cells induced the endoplasmic reticulum (ER) stress and upregulated the unfolded protein response (UPR) pathway in both AML cells and BM stromal cells [132]. Mechanistically, the UPR pathway was upregulated in AML-derived EVs and their transfer into mesenchymal stromal cells and osteoprogenitor cells caused an induction of ER stress and upregulation of UPR in the recipient cells [132]. ER stress-mediated UPR upregulation is known to have protective effect on cancer cells increasing their resistance to chemotherapy [133,134].

The opposite effect of BM stromal cells on AML cells via exosome secretion is less well studied. However, it was reported that exosomes from BM mesenchymal stromal cells transfer miR-7-5p into AML cells and induce apoptosis through inhibition of the PI3K/AKT/mTOR signaling pathway indicative of a protective role of BM-derived exosomes against AML [135].

Several studies have also demonstrated the negative impact of AML cells on normal hematopoietic cells by suppressing hematopoietic cell differentiation and reducing the hematopoietic stem cell population. In this process the development of an inflammatory microenvironment seems to be essential [136]. Particularly, AML cells release exosomes enriched in miR-150 and miR-155, which suppress the differentiation and proliferation of hematopoietic stem/progenitor cells (HSPC) [137]. Additionally, it was demonstrated that AML-EVs could rapidly enter in and induce quiescence to long-term HSC (LT-HSC) via the transfer of miR-1246, which targets the mTOR subunit Raptor [138]. This in turn leads to reduced phosphorylation of the ribosomal protein S6 and impaired protein synthesis in LT-HSC [138]. Finally, it has been demonstrated that AML cells through release of exosomes suppressed the migration capacity of healthy pre-B cells [131].

AML-EVs can affect HSC indirectly by educating the cells of the BM microenvironment to become less supportive of the HSC. Specifically, AML-exosomes, once taken up by stromal cells, induce the expression of *DKK1*, which suppressed normal hematopoiesis and promoted leukemia development [139]. In vitro and murine xenograft studies also supported both a direct and indirect, via BM stromal cells, negative effect of AML-EV on the retention and clonogenicity of HSC [140]. The reported findings related to the role of EVs in AML are summarized in Table 1.

In summary, there is intensive research supporting the reprogramming of the BMM, including the healthy HSC, by the AML cells via the release of EVs to make it more reinforcing and hospitable, inducing the survival of AML cells and their resistance to chemotherapy or targeted therapies.

**Table 1 ijms-23-08827-t001:** Summary of findings related to the role of extracellular vesicles in myeloid malignancies.

Disease	Extracellular Vesicle	Cargo	Association/Effect	References
ET	Platelet MPs	Platelet and endothelial markers (CD61, CD144)	Hypercoagulable state, increased thrombosis risk	[70]
ET	BM exo	circDAP3, circASXL1, circRUNX1	Decreased exo number	[74]
PV	Platelet EVs	Protein diversity, specific DNA microbial signature	Increased EV number, pro-coagulation, inflammation	[76,77]
PMF	Platelet EVs, endothelial cell EVs, erythrocyte EVs	N/A	Increased EV number	[83]
PMF-TN	Plasma EVs	mRNA (miR-361-5p)	CD34^+^ cell survival	[85]
CML	CML-exo	TGFβ1	Apoptosis inhibition	[97]
CML	CML-exo	Amphiregulin (AREG)	Enhanced CML proliferation via BM stroma, increased adhesion to BM stroma	[98]
CML	CML-exo	Ν/A	Polarization of MΦ to tumor-associated MΦ	[99]
CML	CML-exo	Ν/A	Induced angiogenic activity of EC	[102]
CML	BM stromal cell-exo	FGF2	TKI resistance	[103]
MDS	Plasma EVs	miR-1237-3p, U33, hsa_piR_019420, miR-548av-5p	Poor survival	[117]
MDS	MDS EVs	N/A	Suppressed MSC differentiation to OB	[111]
MDS	MSC EVs	miR-10a, miR-132	Altered EV protein expression, increased viability and clonogenicity of CD34^+^ cells	[118]
MDS	MSC EVs	miR-486-5p	Increased DNA damage and mutagenesis of HSC	[120]
AML	AML MVs	TGFβ1	Suppressed NK function	[130]
AML	AML EVs	IGF-IR	Increased proliferation and VEGF expression in MSC	[131]
AML	AML EVs	UPR	ER stress in MSC and osteoprogenitors	[132]
AML	MSC exo	miR-7-5-p	Increased AML apoptosis, inhibition of PI3K/AKT/mTOR	[135]
AML	AML exo	miR-150, miR-155	Suppressed HSPC differentiation and proliferation	[137]
AML	AML EVs	miR-1246	Increased LT-HSC quiescence via Raptor	[138]
AML	AML exo	AML-related coding and non-coding RNAs	Reduced migration of pre-B cells	[131]
AML	AML exo	N/A	Suppressed osteogenesis and normal hematopoiesis via DKK1, leukemia development	[139]

ET = essential thrombocytosis. MPs = microparticles. MVs = micro-vesicles. Exo = exosomes. BM = bone marrow. VEGF = vascular endothelial growth factor. ER = endoplasmic reticulum. UPR = unfolded protein response. EVs = extracellular vesicles. TGFβ1 = transforming growth factor-beta 1. BM = bone marrow. FGF2 = Fibroblast Growth Factor 2. TKI = Tyrosine kinase inhibitor. MSC = Mesenchymal stromal cells. OB = osteoblasts. HSC = Hematopoietic stem cells. HSPC = hematopoietic stem/progenitor cells. LT-HSC = Long-term hematopoietic stem cells. Raptor = Regulatory-associated protein of mTOR. DKK1 = Dickkopf WNT Signaling Pathway Inhibitor 1. MΦ = macrophages.

## 3. Conclusions

The consistency of EVs from patients with myeloid neoplasms differs from healthy individuals with alterations of micro-RNA expression being the best-described difference. Recent studies support a possible role of EVs in the crosstalk between malignant hematopoietic cells and components of the BMM. Particularly, altered micro-RNAs in the secreted EVs promote the activation of growth-related pathways such as AKT/mTOR signaling in malignant myeloid cells and the acquisition of stem cell-like features by them. Moreover, EVs secreted by the malignant cells are responsible for the suppression of healthy hematopoiesis, either by BMM reprogramming or by direct effect on healthy hematopoietic cells. Further mechanistic work is required to discover the pathways associated with the regulation of EVs secretion by malignant and stromal cells. Despite this, a number of agents inhibiting the biogenesis and release of EVs have been found to have anti-cancer activity [141,142,143], but these agents have not been investigated in myeloid neoplasms. Thus, it would be interesting to investigate the efficacy of these drugs in models of these diseases, such as in vitro models of de novo and secondary AML and xenograft models. However, it should be taken into consideration that the secretion of EVs in the BMM can affect healthy hematopoiesis at multiple levels. Thus, evaluating the effect of these agents in healthy CD34+ in the presence and absence of BM stroma will be critical before further investigation.

## Figures and Tables

**Figure 1 ijms-23-08827-f001:**
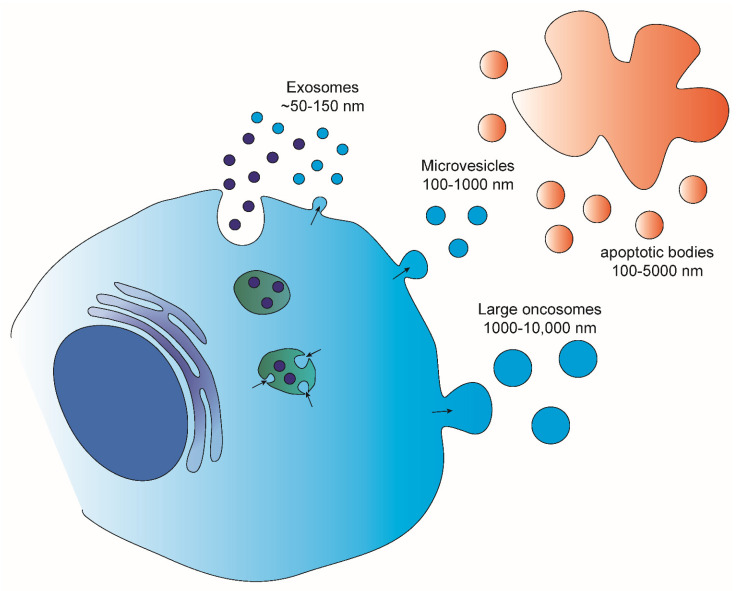
Types, sizes and biogenesis of the main types of extracellular vesicles. Exosomes are mainly derived from the fusion of the multivesicular bodies with the plasma membrane and are between 50–150 nm in size. Apoptotic bodies are bigger (100–5000 nm) and are formed by membrane budding of apoptotic cells. Finally, micro-vesicles (100–1000 nm) and large oncosomes (1000–10,000 nm) are released via the budding of the plasma membrane.

## Data Availability

Not applicable.

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
