# Peer review of "Extracellular Vesicles in Myeloid Neoplasms"

_ijms, 2022, doi:10.3390/ijms23158827_

Round 1
Reviewer 1 Report
A review work entitled ‘’ Extracellular vesicles in myeloid neoplasms ‘’ reveal RNA and protein alterations in EVs isolated from patients with myeloid neoplasms compared to healthy controls. Moreover, mechanistic studies have 19 demonstrated that EVs secreted from the malignant cells or from the re-programmed BMM are implicated in the growth of malignant clones and the suppression of healthy hematopoiesis. These results support the possible role of EVs in the pathogenesis of myeloid neoplasms and warrant further studies to understand their biology better. The results in interesting. However, I provided some comments on the manuscript. Authors should address concerns.
1. The text is written fluently and is clear, however, some typo errors must be corrected.
2. Authors revise the abstract and add the main findings instead of the background.
3. Authors use these articles in sections (PMID: 33302971)
4. Authors should provide informative figures for readers.
5. In Table 1, check references.
6. It is better to expand the conclusion with respect to the advantages and disadvantages of EVs applications.
Author Response
Reviewer 1
- The text is written fluently and is clear, however, sometypo errors must be corrected.
We reviewed our manuscript extensively, identified typos and corrected them.
- Authors revise the abstract and add the main findings instead of the background.
As a response to the reviewer’s comment, we revised our abstract by decreasing the section describing background information and including the main findings related to the effect of extracellular vesicles in the bone marrow microenvironment of patients with myeloid neoplasms (lines: 34 – 40).
- Authors use these articles in sections (PMID:33302971)
As a response to the reviewer’s comment, we have added this article to our introduction section (lines: 66 – 69).
- Authors should provide informative figures for readers.
As a response to the reviewer’s comment, we have created a figure presenting the biogenesis and describing the different subtypes of extracellular vesicles (lines 81, 1177 – 1183).
- In Table 1, check references.
We have reviewed the references in the Table 1 and have ensured that they are correct.
- It is better to expand the conclusion with respect to the advantages and disadvantages of EVs applications.
We thank the reviewer for this comment. As a response we have included a statement in the conclusion supporting that the investigation of EV biogenesis and secretion inhibitors in models of myeloid neoplasms will be promising. However, we have highlighted that careful evaluation of the effects on healthy hematopoiesis is required. Of note, these inhibitors have not been studied in malignant hematopoiesis models so far (lines: 474 – 481).

Reviewer 2 Report
A medline search for papers published in the last 5 years using as keywords “extracellular vesicles/exosomes” and “cancer” identified 8097 hits, 2465 of which are reviews. A good proportion of these papers are published in high impact journals. By contrast the same search using “extracellular vesicles/exosomes” and “myeloproliferative neoplasms” as keywords identified only 39 papers, 39 of which are reviews. Many of these studies come from a single laboratory and, with few exception such as the paper by the Villeval group published in JCI, are published in low-middle impact journals.
The disproportion between the numbers of studies published on extracellular vesicles and cancer and those published in hematopoietic neoplasms indicates that this review is timely and will stimulate the interest to perform further studies to characterize the role played by extracellular vesicles in the progression and resistance to therapy of myeloproliferative neoplasms.
Having said that, it must also be said that, overall, the review was found to be written in generic terms, leaving the impression that the field is still too speculative to have a relevant clinical impact on myeloid neoplasms. It is suggested to reduce the perception of “speculation” by emphasizing the (few) results which have been validated so far, such as 1) numbers of circulating EVs and thrombosis in ET and PV, 2) unicity of the EVs in triple negative Philadelphia-negative MPN, 3) FGF2-containing exosomes as possible driver of TKI resistance in Philadelphia-positive MPN; 4) The correlation of the EV load for four miRNAs and survival in MDS and 5) the suppression of immune-surveillance exerted by EVs from AML. Also, mechanistic relations between EVs content and clinical features require to be more deeply elaborated. Specific suggestions are provided below.
Specific comments
The references include several recent reviews on the role of extracellular vesicles in cancer published in very good journals. Given the enormous number of reviews published on this subject in the last 6 years, it is understood that the authors could not code all of them and had to be picky. For this reason, to reduce the perception of biased quotation, it is suggested to code only the most recent reviews published in the highest journals and to limit those published in less prestigious journals only to those that really make points not covered by others. In this case, however, it would be preferable that the authors code the original paper rather than the review.
Page 2 first line “encapsulated in a lipid bilayer membrane”. This statement, and other similar mentioned afterword, are imprecise and should be revised. The components of the EV are encapsulated in portions of the plasma membrane of the cells from which they derive. Since the plasma membrane contains not only a lipid bilayer but also adhesion receptors and other cell-type specific surface proteins, the proteins embedded in the EV membrane provide the basis for the specificity of the signaling between the cell which have released the EV and the target cell.
Page 2, end of the second paragraph: “to employ EVs as source of biomarkers to follow up disease progression and/or treatment response”. This sentence should be modified to specify the type of cancers in which the use of EVs as biomarkers has been validated. This is important to reduce the perception that the paper reviews an interesting but highly speculative field which had not moved to the clinic as yet.
Page 2, last sentence. It is strongly recommended to include a figure summarizing the different modality of EV generation and the proteins identified so far that regulates their formation.
Page 3: I do not understand this sentence: PV is also a relatively indolent my-133 eloid neoplasm characterized by elevated red blood cell production, but its natural history 134 might be interrupted by thrombotic, fibrotic, or leukemic events, with respective 20-year 135 rates of 26%, 16%, and 4%[54]. Please revise for clarity.
Page 5, first paragraph: the bioactive molecules specifically present in EV that alter the cell behaviour and the nature of the changes induced should be spelled out.
Page 5, second paragraph: “Of note, elevated CD41+ microparticles were positively associated with increased thrombosis risk among ET patients[65]. Importantly, elevated plasma microparticles have 163 been positively associated with increased thrombin generation among ET patients[66]”. These two observations are described disjointly while their mechanistic link should be discussed. It is also important to discuss that they both were made in the same laboratory more than 8 years ago and to clarify whether since then they have been independently confirmed by other laboratories. Refs 67 and 68 discussed later on are reviews and not additional primary data. This is important because these reports are among the few concrete examples of EVs as biomarkers for disease manifestation in MPN available.
Page 5, third paragraph: By contrast the pathobiological role of the content of the EVs exosomes present in the bone marrow of ET patients discussed in this paragraph is counterintuitive. The fact that these EVs have been shown to inhibit maturation of cell lines toward megakaryocytes makes it unclear how they may facilitate the progression of a disease characterized by excessive megakaryocyte proliferation.
Page 5, least paragraph: This paragraph discusses that elevated EVs predicts risk for thrombosis also in PV. This paragraph is overall dry. It requires some more words to detail the difference between the EVs found in ET and PV and to clarify how EVs with at least partially different content may predict risk for thrombosis both in ET and PV. Is there more than one mechanism to trigger thrombosis and the two sets of EVs act at different levels?
Page 6, first paragraph: “The natural history of PMF is characterized by particularly altered cytokine profile 203 in the BM microenvironment, which is directly linked with the excess marrow fibrosis and 204 bone marrow failure[76]. Consistently, circulating EVs are significantly elevated in pa-205 tients with PMF compared to healthy controls[77]”. Why the elevated levels of circulating EV in PMF is consistent with the altered cytokine profile and bone marrow failure observed in this patient? This sentence should be either clarified or deleted. In addition, the authors should specify the criteria used to define EVs generated by red cells versus those generated by platelets in the various studies.
Page 6, second paragraph: “EVs isolated from the plasma of patients with TN disease showed distinct phenotypes and specific mRNA signatures compared to the ones isolated from PMF patients with JAK2 mutation or healthy individuals[80]”. The “distinctive phenotypes” and the “mRNA signatures should be defined. “Additionally, it was demonstrated 223 that EVs derived from patients with TN disease promoted the survival of CD34+ cells 224 from TN patients[80]”. The mechanistic implications of this observation should be discussed. Of note, this is one of the few reports supporting the generic statement made by the authors that EV support the growth of the malignant stem cells. However, since the mutations of the TN CD34+ cells are not known, it is not possible to state whether these EV supported the growth on the normal or of the malignant CD34+ cells.
Page 6, third paragraph: “Consistently, monocytes from PMF patients with JAK2V617F mutation released reduced levels of EVs-linked inflammatory cytokines (IL1β, IL-6, TNFα, IL-10) upon lipopolysaccharides stimulation[81]”. This observation is weak given the fact that in PMF all these proinflammatory cytokines are present at high levels because produced by other cells. The authors should be more critical with the descriptions of the findings present in the literature. It is conceivable that not all the EVs abnormalities found in the patients will have clinical implications.
Page 7, second paragraph: ” CML patients increased the expression of the metalloproteinase MMP9 and the cytokine IL8 by BM stroma cells, which in turn promoted the proliferation and survival of leukemia cells”. This sentence, and moreover the following sentence on Annexin 2, require few words to explain the mechanistic implications of these findings.
Page 7, third paragraph: “BCR-ABL1 oncogene”. Is this as meant or do the authors rather mean: the protein encoded by the BCR-ABL1 oncogene?
Author Response
Reviewer 2
In response to reviewer’s general comment in the revised manuscript we aimed to provide more detailed information on the studies that are published on the role of EVs in myeloid neoplasms, including any functional implications using the original papers in higher impact factors journals when that was possible.
- Page 2 first line “encapsulated in a lipid bilayer membrane”. Thisstatement, and other similar mentioned afterword, are impreciseand should be revised. The components of the EV areencapsulated in portions of the plasma membrane of the cells fromwhich they derive. Since the plasma membrane contains not only alipid bilayer but also adhesion receptors and other cell-type specific surface proteins, the proteins embedded in the EV membraneprovide the basis for the specificity of the signaling between thecell which have released the EV and the target cell.
We have revised this sentence in our revised manuscript (lines: 62-65)
- Page 2, end of the second paragraph: “to employ EVs as source ofbiomarkers to follow up disease progression and/or treatmentresponse”. This sentence should be modified to specify the type ofcancers in which the use of EVs as biomarkers has been validated.This is important to reduce the perception that the paper reviewsan interesting but highly speculative field which had not moved tothe clinic as yet.
As a response to reviewer’s comment we have added information on studies in high-impact journals that have detected specific signatures in exosomes released by melanoma, breast, pancreatic and prostate cancer and suggested them as predictive and/or therapeutic biomarkers. (lines: 94-103)
- Page 2, last sentence. It is strongly recommended to include afigure summarizing the different modality of EV generation and theproteins identified so far that regulates their formation.
As a response to the reviewer’s comment, we have created a figure presenting the biogenesis and describing the different subtypes of extracellular vesicles.
- Page 3: I do not understand this sentence: PV is also a relatively indolent myeloid neoplasm characterized by elevated red blood cell production, but its natural history 134 might be interrupted by thrombotic, fibrotic, or leukemic events, with respective 20-year 135 rates of 26%, 16%, and 4%[5 4]. Please revise for clarity.
We have revised the specific sentence in our revised manuscript (lines: 158 – 162).
- Page 5, first paragraph: the bioactive molecules specificallypresent in EV that alter the cell behaviour and the nature of thechanges induced should be spelled out.
We have revised this specific sentence mentioning the molecules present in the EVs and the cellular behaviors that may be influenced. (lines 184-186)
- Page 5, second paragraph: “Of note, elevated CD41+microparticles were positively associated with increasedthrombosis risk among ET patients[65]. Importantly, elevatedplasma microparticles have 163 been positively associated withincreased thrombin generation among ET patients[66]”. These twoobservations are described disjointly while their mechanistic linkshould be discussed. It is also important to discuss that they bothwere made in the same laboratory more than 8 years ago and toclarify whether since then they have been independently confirmedby other laboratories. Refs 67 and 68 discussed later on arereviews and not additional primary data. This is important becausethese reports are among the few concrete examples of EVs asbiomarkers for disease manifestation in MPN available.
We have revised this paragraph and made sure the findings are presented as continuous (lines 195-203).
- Page 5, third paragraph: By contrast the pathobiological role of thecontent of the EVs exosomes present in the bone marrow of ETpatients discussed in this paragraph is counterintuitive. The factthat these EVs have been shown to inhibit maturation of cell linestoward megakaryocytes makes it unclear how they may facilitatethe progression of a disease characterized by excessivemegakaryocyte proliferation.
We thank the reviewer for bringing up this question. The inhibition of maturation of K562 cells to megakaryocytes may suggest that EVs have the ability to suppress differentiation. This is a process associated with progression of MPN from early stages (such as ET) to more advanced stages such as myelofibrosis and acute leukemia. We have revised our manuscript to clarify this point (lines 208 – 210).
- Page 5, least paragraph: This paragraph discusses that elevatedEVs predicts risk for thrombosis also in PV. This paragraph isoverall dry. It requires some more words to detail the differencebetween the EVs found in ET and PV and to clarify how EVs withat least partially different content may predict risk for thrombosisboth in ET and PV. Is there more than one mechanism to triggerthrombosis and the two sets of EVs act at different levels?
An additional statement on the lack of literature on the underlying mechanisms by which these EV express their pro-coagulant activity and on the comparison between the two diseases was added in the revised manuscript (lines 225-228)
- Page 6, first paragraph: “The natural history of PMF ischaracterized by particularly altered cytokine profile 203 in the BMmicroenvironment, which is directly linked with the excess marrowfibrosis and 204 bone marrow failure[76]. Consistently, circulatingEVs are significantly elevated in pa- 205 tients with PMF comparedto healthy controls[77]”. Why the elevated levels of circulating EVin PMF is consistent with the altered cytokine profile and bonemarrow failure observed in this patient? This sentence should beeither clarified or deleted. In addition, the authors should specifythe criteria used to define EVs generated by red cells versus thosegenerated by platelets in the various studies.
In the revised manuscript we have clarified the above statement where EV are part of an altered bone marrow microenvironment where there is also alteration in the cytokines profile and specified the type of EVs generated by the different cell types. (lines 237-243)
- Page 6, second paragraph: “EVs isolated from the plasma ofpatients with TN disease showed distinct phenotypes and specificmRNA signatures compared to the ones isolated from PMFpatients with JAK2 mutation or healthy individuals[80]”. The“distinctive phenotypes” and the “mRNA signatures should bedefined. “Additionally, it was demonstrated 223 that EVs derivedfrom patients with TN disease promoted the survival of CD34+cells 224 from TN patients[80]”. The mechanistic implications ofthis observation should be discussed. Of note, this is one of thefew reports supporting the generic statement made by the authorsthat EV support the growth of the malignant stem cells. However,since the mutations of the TN CD34+ cells are not known, it is notpossible to state whether these EV supported the growth on thenormal or of the malignant CD34+ cells.
In the revised manuscript more information was given on the experiment however was clarified that the mechanism was not investigated in depth in this study. Additionally, we added an additional statement on why we think this study is important (lines 260-264)
- Page 6, third paragraph: “Consistently, monocytes from PMFpatients with JAK2V617F mutation released reduced levels of EVs-linked inflammatory cytokines (IL1β, IL-6, TNFα, IL-10) uponlipopolysaccharides stimulation[81]”. This observation is weakgiven the fact that in PMF all these proinflammatory cytokines arepresent at high levels because produced by other cells. Theauthors should be more critical with the descriptions of the findingspresent in the literature. It is conceivable that not all the EVsabnormalities found in the patients will have clinical implications
In the revised mechanism it was given more emphasis in the differences observed in the EV-linked versus free cytokines and the potential importance of the EV-linked cytokines which were the only ones restored after ruxolitinib. (lines 267-275)
- Page 7, second paragraph: ” CML patients increased theexpression of the metalloproteinase MMP9 and the cytokine IL8 byBM stroma cells, which in turn promoted the proliferation and survival of leukemia cells”. This sentence, and moreover thefollowing sentence on Annexin 2, require few words to explain themechanistic implications of these findings.
In the revised mechanism few more sentences were added on the cargo of the EV as well as on the mechanism by which Annexin2 acts. (lines 308-316).
- Page 7, third paragraph: “BCR-ABL1 oncogene”. Is this as meantor do the authors rather mean: the protein encoded by the BCR-ABL1 oncogene?
In the revised manuscript this sentence is corrected.

Reviewer 3 Report
The authors described the role of Extracellular vesicles (EVs) in this well written and comprehensive up-to-date study. Clearly, EVs are important for the future treatment of hematological diseases described by authors.
Minor aspects
Please use Italic for Gene name (it is not uniform)
Maybe one Figure or graphical abstract will increased the interest of the readers
The conclusion (beginning) sound like one Introduction, please formulate clear conclusions
Author Response
Reviewer 3
Minor aspects
- Please use Italic for Gene name (it is not uniform)
We have corrected this point
- Maybe one Figure or graphical abstract will increased the interest of the readers
As a response to the reviewer’s comment, we have created a figure presenting the biogenesis and describing the different subtypes of extracellular vesicles (lines 81, 1177-1183).
- The conclusion (beginning) sound like one Introduction, please formulate clear conclusions
As a response to reviewer’s comment we have revised our conclusion by decreasing the proportion of background information and summarizing the main findings related to the micro-RNA alterations in EVs in myeloid neoplasms and the impact of EVs in healthy hematopoiesis (lines: 483 – 490).
